# Development of Immunochromatographic Test Kit for Rapid Detection of Specific IgG4 Antibody in Whole-Blood Samples for Diagnosis of Human Gnathostomiasis

**DOI:** 10.3390/diagnostics11050862

**Published:** 2021-05-11

**Authors:** Penchom Janwan, Pewpan M. Intapan, Lakkhana Sadaow, Rutchanee Rodpai, Hiroshi Yamasaki, Patcharaporn Boonroumkaew, Oranuch Sanpool, Tongjit Thanchomnang, Phuangphaka Sadee, Wanchai Maleewong

**Affiliations:** 1Department of Medical Technology, School of Allied Health Sciences, Walailak University, Nakhon Si Thammarat 80161, Thailand; pair.wu@gmail.com; 2Department of Parasitology, Faculty of Medicine, Khon Kaen University, Khon Kaen 40002, Thailand; pewpan@kku.ac.th (P.M.I.); sadaow1986@gmail.com (L.S.); rutchanee5020@gmail.com (R.R.); hamooploy@gmail.com (P.B.); oransa@kku.ac.th (O.S.); 3Mekong Health Science Research Institute, Khon Kaen University, Khon Kaen 40002, Thailand; tthanchomnang@gmail.com; 4Department of Parasitology, National Institute of Infectious Diseases, Tokyo 162-8640, Japan; hyamasak@niid.go.jp; 5Faculty of Medicine, Mahasarakham University, Maha Sarakham 44000, Thailand; 6Clinical Immunology Unit, Srinagarind Hospital, Faculty of Medicine, Khon Kaen University, Khon Kaen 40002, Thailand; sphuan@kku.ac.th

**Keywords:** human gnathostomiasis, immunochromatographic test kit, point-of-care test, whole-blood sample, serodiagnosis, IgG4 antibody

## Abstract

Human gnathostomiasis is a harmful food-borne zoonosis caused by roundworms of the genus *Gnathostoma*. The parasite can occasionally migrate to the central nervous system, causing life-threatening disease and death. Here, we report a new point-of-care (POC) test kit, the gnathostomiasis blood immunochromatographic test (GB-ICT) kit. The kit is based on recombinant *Gnathostoma spinigerum* antigen and detects specific IgG4 antibody in whole-blood samples (WBSs). The GB-ICT kit showed potentially high diagnostic values with simulated WBSs (*n* = 248), which were obtained by spiking patients’ sera with red blood cells. The accuracy, sensitivity, specificity, and positive and negative predictive values were 95.2%, 100%, 93.8%, 81.5%, and 100%, respectively. Ten WBSs from clinically suspected gnathostomiasis patients were all positive according to the GB-ICT kit, while 10 WBSs from healthy volunteers were negative. The GB-ICT kit is a simple and convenient POC testing tool using finger-prick blood samples: venous blood sampling and serum separation processes are not required. The GB-ICT kit can support clinical diagnosis in remote areas and field settings without sophisticated equipment facilities.

## 1. Introduction

Human gnathostomiasis is a medically important food-borne parasitic zoonosis caused by nematode worms in the genus *Gnathostoma*. Gnathostomiasis occurs predominantly in Asia and the Americas, and in travelers returning from endemic countries. The infection is mainly acquired by ingestion of raw or undercooked freshwater fish, amphibians, reptiles, birds and mammals, all of which are known to harbor advanced third-stage larvae of *Gnathostoma* species [1,2,3,4,5,6]. Gnathostomiasis patients usually present with intermittent cutaneous migratory swellings and creeping eruption. However, *Gnathostoma* worms can occasionally migrate to the viscera, eye, and central nervous system, causing serious disease and sometimes death [3,7,8,9,10]. Definitive diagnosis of gnathostomiasis can be made by recovery of the migrating larvae from the human body, but this direct detection is extremely difficult. Therefore, in practice diagnosis of gnathostomiasis is based on clinical features, peripheral blood eosinophilia, history of ingesting undercooked parasite-contaminated foods and of living in or traveling to endemic regions, and serological tests [3,4,5,6]. Antibody-detection methods reported for serodiagnosis of human gnathostomiasis include enzyme-linked immunosorbent assays (ELISA) [11,12,13,14] and immunoblot assays [15,16,17,18] using native *Gnathostoma spinigerum* antigens. These had diagnostic sensitivities and specificities ranging from 47.4% to 100% and 69.9% to 100% for total IgG detection [11,12,13,15,18]. Corresponding values for detection of IgG4 were 75.0% to 100%, 93.9% to 100% [16,17]. However, the use of native antigens requires collection of parasites from their freshwater fish hosts, in particular, eels naturally infected with *G. spinigerum* [12,13,14,15,16] and/or maintenance of the parasite’s life cycle using experimental animals [11,17,18]. Such approaches yield only small amounts of antigen. Currently, two types of recombinant antigens, namely recombinant matrix metalloproteinase (rMMP) [19] and Gslic18 protein (rGslic18) [20], are available as diagnostic antigens: both show high sensitivity and specificity for detection of total IgG antibody in the serodiagnosis of human gnathostomiasis. These recombinant antigens have the potential to replace native parasite antigens with a stable mass-production system. The rMMP protein has been used in an immunoblot assay and dot-ELISA to detect anti-*Gnathostoma* IgG antibodies in human sera [19,21,22], with 100% sensitivity and specificities ranging from 94.7% to 100%. However, these conventional assays are rather complicated and time-consuming, requiring costly and sophisticated equipment and well-trained analysts (or medical technologists). A user-friendly and rapid platform is needed for point-of-care (POC) diagnosis in resource-limited settings. The immunochromatographic test (ICT) or lateral flow immunoassay (LFIA) provides such a platform. This combines a chromatographic system with immunochemical reactions for specific and sensitive detection [23,24]. Recently, an ICT platform coupled with a rGslic18 protein (the KAN gnathostomiasis kit) has been developed for the rapid diagnosis of human gnathostomiasis based on total IgG antibody detection with high sensitivity (93.8%) and specificity (97.0%) [20]. However, all of the above-mentioned assays require serum samples and cannot use whole-blood samples (WBSs). In this study, we describe a new ICT kit, named the gnathostomiasis blood immunochromatographic test (GB-ICT) kit, for detection of specific IgG4 antibody in WBSs using rGslic18 as the serodiagnostic antigen to diagnose human gnathostomiasis. Specific IgG4 antibody has been shown to have excellent diagnostic values for serodiagnosis of gnathostomiasis [16,17].

## 2. Materials and Methods

### 2.1. Clinical Samples

Two hundred and forty-eight individual human sera from the serum bank at the Parasitology Department of Khon Kaen University were used in this study, 40 of which were from healthy volunteers who were free of any intestinal parasite infections at the time of blood collection according to examination of stool using the formalin ethyl acetate concentration technique [25]. Fifty-three samples were from gnathostomiasis patients, which included samples from seven parasitologically proven cases and 46 from patients showing clinical symptoms of suspected gnathostomiasis with a history of eating food possibly contaminated with *Gnathostoma* infective larvae and were positive according to the KAN gnathostomiasis rapid ICT kit [20]. The remaining 155 serum samples were from patients with other parasitic diseases, including angiostrongyliasis cantonensis (*n* = 13 cases), trichinellosis spiralis (*n* = 14), strongyloidiasis (*n* = 15), capillariasis philippinensis (*n* = 10), ascariasis (*n* = 10), trichuriasis (*n* = 10), hookworm infections (*n* = 10), paragonimiasis heterotremus (*n* = 13), fascioliasis gigantica (*n* = 10), opisthorchiasis viverrini (*n* = 15), sparganosis (*n* = 5), taeniasis saginata (*n* = 15), and cysticercosis (*n* = 15). These infections had been diagnosed parasitologically, except for cysticercosis patients, who were diagnosed using computerized tomography and serological methods [26]. Some of the patients with trichinellosis spiralis (13/14), angiostrongyliasis cantonensis (6/13), and fascioliasis gigantica (7/10) had been diagnosed on the basis of clinical symptoms, histories of consuming food sources associated with infection and specific serological tests [27,28,29]. Pooled positive and negative reference sera were prepared by mixing equal volumes of sera from 10 gnathostomiasis patients and 10 healthy volunteers, respectively. These pooled positive and negative sera were used as control sera to assess the between-day precision of the ICT kit.

EDTA anti-coagulated WBSs of 10 clinically suspected gnathostomiasis patients who had returned a positive serological test [20] and of 10 healthy volunteers whose stool samples were not found from any intestinal parasitic infection [25] were also used.

### 2.2. Preparation of Simulated WBSs

Simulated WBSs were prepared to determine the diagnostic values of the new ICT kit. Red blood cells (RBCs) were added to sera. We employed RBCs from leftover WBS, blood group O in citrate phosphate dextrose adenine-1 anticoagulant, from healthy donors of the Central Blood Bank, Faculty of Medicine, Khon Kaen University. The WBS (0.5 mL) was centrifuged at 13,200× *g* for 10 min at 4 °C and plasma removed. The packed RBCs were washed three times with phosphate-buffered saline (PBS), pH 7.4, by centrifugation at 13,200× *g* for 10 min at 4 °C. The packed RBCs were re-suspended in PBS and the suspension divided into 10 µL aliquots, which were then centrifuged at 13,200× *g* for 10 min at 4 °C. The supernatant (6.5 µL) was discarded and the remaining packed RBCs (3.5 µL) were used for preparation of simulated WBSs. Before performing the experiment, these packed RBCs were re-suspended in serum from a frozen bank as above (6.5 µL; to restore the simulated WBS to normal levels of human blood components) or kept at 4 °C for further experiments.

### 2.3. Preparation of Recombinant Antigen

The recombinant antigen (rGslic18) was prepared according to the method previously reported [20]. Briefly, a clone of rGslic18-pQE-31-*Escherichia coli* XL-1 Blue in glycerol stock was used as a starter for protein expression. Expression of N-terminal-fused His-tag rGslic18 was induced by adding 1 mM isopropyl 1-thio-β-d-galactopyranoside at 37 °C for 18 h. The rGslic18 protein expressed as an insoluble protein was solubilized using urea solution (8 M urea, 0.1 M NaH_2_PO_4_, 0.01 M Tris-HCl, pH 8.0). The supernatant containing rGslic18 protein was then purified by affinity chromatography using a nickel-charged column installed on an ÄKTA purifier fast-protein liquid-chromatography system (GE Healthcare, Uppsala, Sweden) at a flow rate of 1.0 mL/min. The purity of the rGslic18 protein was confirmed by sodium dodecyl sulfate-polyacrylamide gel electrophoresis stained with Coomassie Brilliant Blue reagent. Fractions containing the rGslic18 protein of interest (14 kDa) were pooled and diluted with distilled water (final concentration of 6 M urea) before adding to a Nanosep^®^ centrifugal device with Omega membrane 3K (Pall Corporation, Cortland County, NY, USA) for protein concentration. The concentration of the purified rGslic18 protein was determined using the Bradford Protein Assay (Bio-Rad Laboratories, Inc., Hercules, CA, USA) and the protein was stored at −70 °C before use.

### 2.4. Preparation of An Immunochromatographic Device

Two types of sample pads (Millipore C048 (Millipore, Burlington, MA, USA) and Cytosep 1660 (Pall Gelman Sciences, Champs-sur-Marne, France)) and four surfactant types (Triton X-100 (Panreac Química SA, Barcelona, Spain), Triton X-405 (Panreac Química SA), Tween 20 (Merck, Darmstadt, Germany) and Tween 80 (Merck) dissolved in running buffer (25 mM Tris-HCl, pH 8.0)) were tested. After testing, sample pad, buffer, and preparation of an antigen-immobilized membrane was as follows: the purified rGslic18 protein and anti-mouse IgG (Lampire Biological Laboratories, Pipersville, PA, USA) were diluted to concentrations of 2 mg/mL and 1 mg/mL, respectively, and sprayed onto a nitrocellulose membrane (Sartorius Stedim Biotech SA, Goettingen, Germany) using the XYZ3210 Dispense Platform (BioDot, Irvine, CA, USA) in a 1-mm-wide line to serve as the test line (T) and control line (C), respectively, at a flow rate of 0.1 μL/mm. Preparation of the conjugate pad was as follows; mouse anti-human IgG4 (6 μg/mL) conjugated with colloidal gold particles 40 nm in diameter was sprayed onto a glass microfiber filter (Whatman Schleicher and Schuell, Dassel, Germany) at a flow rate of 1 μL/mm. The antigen-immobilized membrane and conjugate pad were then dried at 37 °C for 2 h, and kept in a desiccator at room temperature until use. The conjugate pad, Millipore C048 sample pad, and absorbent pad were assembled with the antigen-immobilized membrane on a laminate card. The assembled sheet was cut into strips 5 mm in width using a guillotine cutter (BioDot, Irvine, CA, USA). An assembled strip is shown in Figure 1a. The strip was covered with a plastic housing (Figure 1b) (Adtec Inc., Oita, Tokyo, Japan) and then packaged in an aluminum foil pouch with desiccant. The gnathostomiasis ICT kit (GB-ICT kit) consisted of an immunochromatographic device, a dropper bottle of buffer (0.1% Triton X-405 in 25 mM Tris-HCl, pH 8.0) for diluting the blood sample and to facilitate chromatography, an instruction manual and a reference card to assist interpretation of the color intensity at the test line. The immunochromatographic device and buffer were stored at 4 °C until further use.

### 2.5. Immunochromatographic Testing Method

The samples to be tested, the immunochromatographic device and the buffer all need to be equilibrated to room temperature before use. In the testing process, 15 μL of optimum diluted (simulated) WBSs was mixed well and applied in the sample well, followed immediately by 2 drops of buffer in the buffer well. The appearance of bands at the C- and T-lines should be observed in intervals from minute to minute to 20 min, and at the optimum time point of 15 min, was used. The presence of the bands at both the C- and T-lines indicates a positive result, while the absence of a band at the T-line is a negative result (Figure 2a). Absence of a band at the C-line indicates technical failure of the test. The intensity of the band at T-line was estimated visually according to the reference card (Figure 2b, with level 1 as the cutoff level). The standard diagnostic indices including accuracy, sensitivity, specificity, and positive and negative predictive values were calculated as previously described [30]. Variables measured were the number of true positives (TP), number of true negatives (TN), number of false positives (FP), and number of false negatives (FN). Test accuracy, the proportion of all tests that gave a correct result, was defined as (TP + TN)/number of all tests. Sensitivity was calculated as TP/(TP + FN), specificity was calculated as TN/(TN + FP), the positive predictive value was calculated as TP/(TP + FP) and negative predictive value was calculated as TN/(TN + FN).

## 3. Results

### 3.1. GB-ICT Kit Optimization and Development

To optimize the LFIA, we evaluated the important parameters to increase the test performance. These included the types of sample pad, the surfactants in running buffer, and dilution of WBS. Millipore C048 and Cytosep 1660 were tested for their suitability as sample pads. The Cytosep 1660 pad did not trap hematocytes effectively, causing leakage of blood into the test strip membrane and interference with result interpretation. The Millipore C048 pad consistently performed better than the Cytosep 1660. In tests of surfactants, the highest intensity of the C- and T-lines was clearly obtained with running buffer that contained 0.1% Triton X-405. Various dilutions (undiluted, 1:10, 1:20, 1:30, 1:40, 1:50, and 1:60) of WBS were titrated. The optimum dilution was found to be 1:30 (1 µL of sample:1 drop (approximate 30 µL) of buffer).

To determine the cutoff level of the GB-ICT kit, cutoff levels of 0.5 and 1 were initially compared. At level 0.5, one sample out of 40 healthy volunteers and all 53 gnathostomiasis patients were positive, while at a cutoff level of 1, all 40 healthy volunteers were negative and all 53 gnathostomiasis patients were positive. The latter cutoff level was therefore selected.

### 3.2. Evaluation of the GB-ICT Kit for Diagnosis of Human Gnathostomiasis

The GB-ICT kit was evaluated using simulated WBSs from healthy volunteers, gnathostomiasis patients, and patients with other parasitic diseases (Table 1 and Appendix A). None of the 40 simulated WBSs from healthy persons showed positive results. Seven simulated WBSs (7/7) from the proven gnathostomiasis patients and 46 from the clinically suspected gnathostomiasis patients (46/46) yielded positive results using a cutoff value of 1.0. Some cross-reactivities were observed in simulated WBSs of angiostrongyliasis cantonensis (1 of 13), trichinellosis spiralis (3 of 14), strongyloidiasis (1 of 15), paragonimiasis heterotremus (2 of 13), opisthorchiasis viverrini (1 of 15), taeniasis saginata (2 of 15), and cysticercosis (2 of 15). The diagnostic accuracy, sensitivity, specificity, and positive and negative predictive values for the GB-ICT were 95.2%, 100%, 93.8%, 81.5%, and 100%, respectively. Additionally, 10 real WBSs from clinically suspected gnathostomiasis patients were all positive according to the ICT kit, while 10 real WBSs from healthy volunteers were all negative.

The detection limit of the kit was found to be stable when stored in an aluminum foil bag containing silica gel desiccant for 12 months at ambient temperature (25 °C) and for 18 months at 4 °C.

## 4. Discussion

The diagnosis of human gnathostomiasis is based on clinical features, history of ingesting raw or undercooked meat of the second intermediate fish in endemic areas, worm recovery and serological techniques. Many attempts have been made to establish a specific serodiagnostic assay system using native parasite or recombinant antigens, including ELISA [11,12,13,14], immunoblot assays [15,16,17,18] and recently the rapid ICT platform [20]. However, to our best knowledge, all of these require serum samples and cannot make use of WBSs. In this study, the novel GB-ICT kit was successfully developed in a laboratory setting for detection of specific IgG4 antibody in WBSs. The GB-ICT strip requires only 1 μL of a WBS for detection and the results can be assessed within 10–15 min by the naked eye. Ten real WBSs from clinically suspected gnathostomiasis cases were all positive according to the kit. The GB-ICT kit showed high sensitivity (100%) and specificity (93.8%) against simulated WBSs. The difference in diagnostic values between the KAN gnathostomiasis kit (93.8% sensitivity and 97.0% specificity) [20] and the GB-ICT kit may be partially due to the different types and number of sera used, the different phases of the infection when samples were collected, and the different types of specific immunoglobulin detected. Cross-reactions in the GB-ICT kit were observed among some samples from patients infected with other species of parasites. These samples were collected from patients in areas endemic for gnathostomiasis who might have had a previous history of subclinical infection with *Gnathostoma* worms. Presence of occasional cross reactions does not cause a major problem in the clinical setting, because infections with these other parasite species usually present with different clinical features from those of gnathostomiasis [31,32,33,34]. IgG4 antibody is associated with chronic antigenic stimulation provided by helminth infection [35,36,37,38]. Human gnathostomiasis is caused by larvae of *Gnathostoma* spp., which migrate aimlessly in the human body for a long time, likely continuously stimulating antibody production [4]. Targeting IgG4 antibody gave our kit high sensitivity for serodiagnosis of human gnathostomiasis, agreeing with a previous study using an immunoblot technique that also targeted IgG4 [16].

At present, several ICT kits are available for assaying analytes in WBS. Most of these kits use a plasma-separation membrane for trapping hematocytes and only plasma migrates into a test strip membrane [23,39,40,41]. However, in our preliminary experiments, the plasma-separation membrane (Cytosep 1660) did not work for trapping hematocytes as expected. To overcome this problem, we used Triton X-405 mixed with the buffer in order to lyse hematocytes before the application of the sample and to enhance the lateral-flow performance of the gnathostomiasis ICT kit, leading to clear observation on the membrane strip. A similar method to lyse hematocytes in WBS by applying detergent has been reported [42,43]. Our GB-ICT kit can also use hemolyzed blood sample, serum, and plasma samples (data not shown). Moreover, our system used the purified rGslic18 of *G. spinigerum* [20] as an antigen, which allows a never-ending source and stable mass production system. This kit provides a straightforward and efficient system that can be easily implemented for laboratory diagnostic purposes. Limitations of this study should be acknowledged. First, we only had a small number of WBSs from parasitologically proven cases for diagnostic evaluation: additional samples are required to optimize this device for use in clinical diagnosis. Second, false negative results may be obtained in acute gnathostomiasis cases in which there has not been adequate time to mount a specific and detectable antibody response.

## 5. Conclusions

No POC diagnostic kit for detection of anti-*Gnathostoma* antibody using WBSs has been previously available. Our ICT kit is the first with the ability to detect specific IgG4 antibody in WBSs for diagnosis of human gnathostomiasis. This GB-ICT kit showed high sensitivity and specificity in simulated WBSs obtained by spiking patients’ serum with red blood cells. A further advantage of this POC kit is that it can make use of finger-prick blood samples: drawing of venous blood and serum separation processes are not required. The kit is appropriate for supporting clinical diagnosis in the field and in remote areas where sophisticated equipment is not available.

## Figures and Tables

**Figure 1 diagnostics-11-00862-f001:**
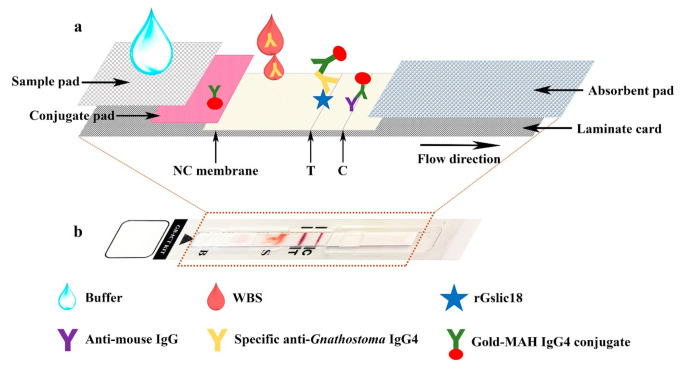
Schematic illustration of the test strip showing the components and how it works (**a**). The whole-blood sample (WBS) containing specific anti-*Gnathostoma* IgG4 antibody is added onto the nitrocellulose (NC) membrane, then it is captured by recombinant Gslic18 (rGslic18) antigen to form an antigen-antibody (Ag-Ab) complex. Following the application of a buffer onto the sample pad that overlaps with the conjugate pad, the mouse anti-human (MAH) IgG4 conjugated with colloidal gold is captured by the Ag-Ab complex on the test line (T), resulting in a red band. The gold-MAH IgG4 conjugate is captured by anti-mouse IgG antibody on the control line (C), resulting in a red band. Photograph of the closed plastic housing containing the test strip (**b**). C, control line; T, test line; S, sample well; B, buffer well.

**Figure 2 diagnostics-11-00862-f002:**
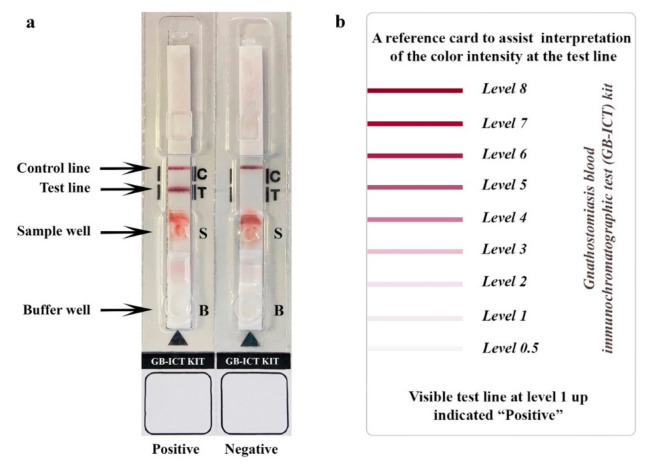
The immunochromatographic test kit for diagnosis of human gnathostomiasis. Representative images of ICT test strip outcomes (**a**) on which positive (**left**) and negative (**right**) results are shown. C, control line; T, test line; S, sample well; B, buffer well. The intensity of test line was visually estimated according to the reference card (**b**).

**Table 1 diagnostics-11-00862-t001:** Types of simulated whole-blood samples (WBSs) examined and diagnostic results of the gnathostomiasis blood immunochromatographic test (GB-ICT) kit.

Type of Simulated WBSs	Number Positive/Total Number
Healthy volunteers	0/40
Confirmed gnathostomiasis	7/7
Clinically suspected gnathostomiasis	46/46
Angiostrongyliasis cantonensis	1/13
Trichinellosis spiralis	3/14
Strongyloidiasis	1/15
Capillariasis philippinensis	0/10
Ascariasis	0/10
Trichuriasis	0/10
Hookworm infection	0/10
Paragonimiasis heterotremus	2/13
Fascioliasis gigantica	0/10
Opisthorchiasis viverrini	1/15
Sparganosis	0/5
Taeniasis saginata	2/15
Cysticercosis	2/15
Total number of tests	248
Numbers of true positive, true negative, false positive, and false negative results	53, 183, 12, 0
Accuracy (%) [95% CI]	95.2 [91.7–97.5]
Sensitivity (%) [95% CI]	100 [93.3–100]
Specificity (%) [95% CI]	93.8 [89.5–96.8]
Positive predictive value (%) [95% CI]	81.5 [70.0–90.1]
Negative predictive value (%) [95% CI]	100 [98.0–100]

CI, confidence interval.

## Data Availability

All data generated or analyzed during this study are included in this published article and its supplementary information files. The raw data are available from the corresponding author on reasonable request.

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
