# Peer review of "Development of Immunochromatographic Test Kit for Rapid Detection of Specific IgG4 Antibody in Whole-Blood Samples for Diagnosis of Human Gnathostomiasis"

_diagnostics, 2021, doi:10.3390/diagnostics11050862_

Round 1
Reviewer 1 Report
The authors present their development of new immunochromatographic test for diagnosis of human gnathostomiasis and demonstrate its efficiency using 248 whole-blood samples. The manuscript accords to basic demands of the Diagnostics journal, but needs in some revisions before its recommendation for publication:
- The authors present in the Introduction a row of earlier developed immunotechniques for serological diagnosis of gnathostomiasis. However, these data are not characterized in terms of types of specific immunoglobulins detected. Did the predecessors detect all IgG or IgG4? What are the reasons to prefer the assay of specific IgG4 antibodies? The authors give only a short statement at lines 274-275 about it without detailed and quantitative characterization of previously obtained data confirming advantages of IgG4 testing.
- The centrifugation regimes (lines 112, 114) should be characterized as «g» instead of «rpm», as well as the «rpm» conditions may be reproduced only with the use of the same rotor.
- The protocol presented in the Section 2.2 («Preparation of simulated WBSs») should be accomplished by a reference to a prototype publication. Are the used volumes (10 mkL, 3,5 mkL, 6.5 mkL) taken from the prototype or specifically chosen in the presented study? In the second case the choice needs in additional comments.
- Lines 155-156: «buffer for diluting the blood sample and to facilitate chromatography». The composition of this buffer should be given in the Section 2.4.
- Line 164: «The appearance of bands at the C- and T-lines should be observed within 15 min.» The choice of this time should be grounded.
- The Results section considers only the approbation of the ready tests for clinical samples. By this way, the process of the immunochromatographic test kit development is out of the manuscript. It is recommended to present here key experiments and decisions at the state of test strip development. How concentrations of reactants and media for their application were chosen? The comments at lines 276-287 should be also divided to the description of experiments (Section 3) and the following short opinion (Section 4).
- The authors describe a reference card with 9 levels of coloration for multi-level estimation of the assay results. However, all interpretations of experimental data are based on qualitative «yes-no» estimation. So the authors should either exclude the consideration of a reference card and levels of coloration from the manuscript or present additional data that were obtained using these opportunities.
- The authors state that low coloration (level 0.5) should be interpreted as negative result, but this decision is not grounded. It will be useful to demonstrate tests according to level 1 and to level 0.5 and justify the reasons to consider the coloration with level 0.5 as negative result.
Author Response
Point by point response to Comments and Suggestions for Authors of Reviewer 1
The authors present their development of new immunochromatographic test for diagnosis of human gnathostomiasis and demonstrate its efficiency using 248 whole-blood samples. The manuscript accords to basic demands of the Diagnostics journal, but needs in some revisions before its recommendation for publication:
Reply: We would to thank the reviewer for your kind suggestion. For revised text line number, please see in “Track change” function in Microsoft Word, no mark up.
1. The authors present in the Introduction a row of earlier developed immunotechniques for serological diagnosis of gnathostomiasis. However, these data are not characterized in terms of types of specific immunoglobulins detected. Did the predecessors detect all IgG or IgG4? What are the reasons to prefer the assay of specific IgG4 antibodies? The authors give only a short statement at lines 274-275 about it without detailed and quantitative characterization of previously obtained data confirming advantages of IgG4 testing.
Reply: We modified and added the sentence in introduction part, page 2, lines 53 to 80 and discussion part, page 7, lines 293 to 299.
2. The centrifugation regimes (lines 112, 114) should be characterized as «g» instead of «rpm», as well as the «rpm» conditions may be reproduced only with the use of the same rotor.
Reply: We modified as suggested, please see revised manuscript, page 3, lines 115, 117, and 118.
3. The protocol presented in the Section 2.2 («Preparation of simulated WBSs») should be accomplished by a reference to a prototype publication. Are the used volumes (10 mkL, 3,5 mkL, 6.5 mkL) taken from the prototype or specifically chosen in the presented study? In the second case the choice needs in additional comments.
Reply: Preparation of simulated WBSs by adding 6.5 mkL of serum sample which estimated by the equivalence to restore the simulated WBS to normal levels of human blood components. We modified the sentence as suggested, please see page 3, lines 121 to 122.
4. Lines 155-156: «buffer for diluting the blood sample and to facilitate chromatography». The composition of this buffer should be given in the Section 2.4.
Reply: We added as suggested. We added “(0.1% Triton X-405 in 25 mM Tris-HCl, pH 8.0)”, please see revised manuscript, page 4, lines 166 to 167.
5. Line 164: «The appearance of bands at the C- and T-lines should be observed within 15 min.» The choice of this time should be grounded.
Reply: The appearance of bands at the C- and T-lines should be observed in intervals from minute to minute to 20 min and at the optimum time point of 15 min was used, please see revised manuscript page 4, lines 177 to178.
6. The Results section considers only the approbation of the ready tests for clinical samples. By this way, the process of the immunochromatographic test kit development is out of the manuscript. It is recommended to present here key experiments and decisions at the state of test strip development. How concentrations of reactants and media for their application were chosen? The comments at lines 276-287 should be also divided to the description of experiments (Section 3) and the following short opinion (Section 4).
Reply: We modified as suggested. For better understanding, we modified in Section 2.4 «Preparation of an immunochromatographic device», page 3, lines 145 to 149 and Section 3.1 «GB-ICT kit optimization and development», page 4, lines 191 to 201. The short opinion was already clarified in Section 4 «Discussion», page 7, lines 300 to 308.
7. The authors describe a reference card with 9 levels of coloration for multi-level estimation of the assay results. However, all interpretations of experimental data are based on qualitative «yes-no» estimation. So the authors should either exclude the consideration of a reference card and levels of coloration from the manuscript or present additional data that were obtained using these opportunities.
Reply: The aim of remaining a reference card of coloration for multi-level estimation is for new laboratory setting of gnathostomiasis diagnosis in other endemic area. The type and numbers of sera samples variations may be effected the cutoff level. This reference card is useful for decision of other clinicians or scientists who interest to apply this ICT kit.
8. The authors state that low coloration (level 0.5) should be interpreted as negative result, but this decision is not grounded. It will be useful to demonstrate tests according to level 1 and to level 0.5 and justify the reasons to consider the coloration with level 0.5 as negative result
Reply: We added as suggested. Please see Section 3.1 «GB-ICT kit optimization and development», pages 4 to 5, lines 202 to 206.
Finally, we would like to thank the reviewer very much. Your comments are supportive and helpful.
Reviewer 2 Report
A new ICT kit (Gna thostomiasis Blood Immunochromatographic Test kit,GB-ICT) with 15 min reacting time , for detection of specific IgG4 antibody in WBSs using rGslic18 as the serodiagnostic antigen to diagnose human gnathostomiasis.
Strong
1. 15 min reacting time, only 1 μL of a simple for detection
2. simple, convenient and easy to implement and expected to provide reliable diagnostic results
3.This kit can support clinical diagnosis in the field and in remote areas.
Weak
1. Is the color of the bands on the strip from photoluminescence reaction?
Why the sample well have color?
2. Can you show the detail data of the diagnostic accuracy, sensitivity, specificity, and positive and negative predictive values?
explain the possibility of false negative?
3. Is the WBS diluted process need to mix very well?
Is the different diluted ratio can change the result of cutoff level?
Author Response
Point by point response to Comments and Suggestions for Authors of Reviewer 2
A new ICT kit (Gna thostomiasis Blood Immunochromatographic Test kit,GB-ICT) with 15 min reacting time , for detection of specific IgG4 antibody in WBSs using rGslic18 as the serodiagnostic antigen to diagnose human gnathostomiasis.
Reply: We would to thank the reviewer for your kind suggestion. For revised text line number, please see in “Track change” function in Microsoft Word, no mark up.
Strong
1. 15 min reacting time, only 1 μL of a simple for detection
2. simple, convenient and easy to implement and expected to provide reliable diagnostic results
3. This kit can support clinical diagnosis in the field and in remote areas.
Reply: We thank for you suggested.
Weak
1. Is the color of the bands on the strip from photoluminescence reaction?
Reply: The color of the bands on the strip is not from photoluminescence reaction but from visible reaction of gold particles (Figure 1, page 6). The color of band is principled by mouse anti-human IgG4 conjugated with colloidal gold particles reacted with IgG4 human antibody at T-line and reacted with anti-mouse IgG antibody at C-line.
Why the sample well have color?
Reply: This is the remnant of hemolysate of whole blood samples, however, this is not affected the T-and C-lines appearances.
2. Can you show the detail data of the diagnostic accuracy, sensitivity, specificity, and positive and negative predictive values?
Reply: We provide as suggested, please see revised manuscript, page 4, lines 184 to 189 and revised Table 1, page 5.
explain the possibility of false negative?
Reply: We provide as suggested, please see revised manuscript, page 7, lines 315 to 317.
3. Is the WBS diluted process need to mix very well?
Reply: Yes, for better understanding, we modified the sentence to “…was mixed well and applied…” Please see revised manuscript, page 4, line 175.
Is the different diluted ratio can change the result of cutoff level?
Reply: Yes, the different diluted ratios may change the result of the cutoff level. But the new experiment on optimization and development need to be reevaluated.
Finally, we appreciate the reviewer very much for the kind suggestions. Your comments are supportive and helpful.
Round 2
Reviewer 1 Report
The manuascript has been successfully revised and now may be published